# Pharmacogenetic Profiling of Genes Associated with Outcomes of Chemotherapy in Omani Healthy Controls

**DOI:** 10.3390/genes16050592

**Published:** 2025-05-17

**Authors:** Nahad Al-Mahrouqi, Nada Al Shuaili, Shoaib Al-Zadjali, Anoopa Pullanhi, Hamida Al-Barwani, Aida Al-Kindy, Hadeel Al-Sharqi, Khalid Al-Baimani, Mansour Al-Moundhri, Bushra Salman

**Affiliations:** 1Research Laboratories, Sultan Qaboos Comprehensive Cancer Care & Research Centre, University Medical City, Muscat 123, Oman; n.almahrouqi@cccrc.gov.om (N.A.-M.); n.alshuaili@cccrc.gov.om (N.A.S.); s.alzadjali@cccrc.gov.om (S.A.-Z.); a.pullanhi@cccrc.gov.om (A.P.); h.albarwani@cccrc.gov.om (H.A.-B.); 2Clinical Trials Department, Sultan Qaboos Comprehensive Cancer Care & Research Centre, University Medical City, Muscat 123, Oman; a.alkindy@cccrc.gov.om; 3Pharmacy Department, Sultan Qaboos Comprehensive Cancer Care & Research Centre, University Medical City, Muscat 123, Oman; h.alsharqi@cccrc.gov.om; 4Department of Medical Oncology, Sultan Qaboos Comprehensive Cancer Care & Research Centre, University Medical City, Muscat 123, Oman; k.albaimani@cccrc.gov.om (K.A.-B.);; 5Pharmacy Department, Omani National Hematology and Bone Marrow Transplant Center, University Medical City, Muscat 123, Oman

**Keywords:** population pharmacogenetics, fluoropyrimidines, next-generation sequencing, *DPYD*, *MTHFR*, *ABCC4*, *UPB1*

## Abstract

Background/Objectives: Pharmacogenomic screening plays a crucial role in optimizing chemotherapy outcomes and minimizing toxicity. Characterizing the baseline distribution of genetic variants in specific populations is essential to inform the prioritization of drug–gene combinations for clinical implementation. The objective of this study was to investigate the distribution of pharmacogenetic variants in 36 genes related to the fluoropyrimidine (FP) pathway among healthy Omani individuals, forming a foundation for future studies in cancer patients receiving FP-based chemotherapy. Methods: Ninety-eight healthy Omani participants aged ≥18 years were recruited at the Sultan Qaboos Comprehensive Cancer Care and Research Center. Whole-blood samples were collected, and genomic DNA was extracted. Targeted next-generation sequencing was performed using a custom Ion AmpliSeq panel covering coding exons and splice-site regions of 36 genes involved in FP metabolism and response. Results: A total of 999 variants were detected across the 36 genes, with 63.3% being heterozygous. The *ABCC4* gene had the highest mutation frequency (76 mutations), while *DHFR* and *SMUG1* had the lowest (<10 mutations). In *DPYD*, four functionally significant variants were found at frequencies ranging from 1 to 8.2% of the population. Missense mutations were also observed in *MTHFR* and *UGT1A1*. Three actionable variants in *DPYD* and *MTHFR*, associated with 5-fluorouracil and/or capecitabine response, were identified. Additionally, 27 novel single-nucleotide polymorphisms of unknown clinical significance were detected. Conclusions: This study reveals key pharmacogenetic variants in the Omani population, underscoring the importance of integrating pharmacogenomic testing into routine care to support safer, more personalized chemotherapy in the region.

## 1. Introduction

Cancer remains one of the leading causes of morbidity and mortality worldwide. The World Cancer Research Fund reports that more than 18 million new cases were diagnosed globally in 2022 [1]. Administration of anti-cancer agents is the most widely used treatment modality [2]. Significant inter-individual variability exists in the efficacy and safety of anti-cancer drugs, largely due to genetic differences affecting the pharmacodynamics and pharmacokinetics of these anti-cancer medications [2]. Pharmacogenomics, which is the study of the association between genetic variations at the individual or population level and the variability in drug efficacy, toxicity, and overall therapeutic outcomes, has the potential to provide an informed decision on the appropriate choice and dosage of medications [3]. In oncology, increasing evidence shows that inherited germline genetic variations play a key role in cancer risk and medication response in addition to the acquired somatic mutations. Several drug–gene pairs have been validated, showing substantial effects [4].

One of the most quoted examples in oncology is the dihydropyrimidine dehydrogenase (*DPYD*) polymorphism effect on fluoropyrimidines [4]. The fluoropyrimidine (FP) 5-fluorouracil (5-FU) and its prodrug, capecitabine, are extensively used in the treatment of several malignancies, such as gastrointestinal, breast, and head and neck cancers. They work by inhibiting the enzyme thymidine synthase, consequently interfering with DNA and RNA replication and repair [5]. They have comparable efficacies, but their toxicity profiles differ significantly despite sharing the same mechanism of action and metabolic pathways (Figure 1) [6,7]. Up to 30% of patients experience severe adverse events, and 0.5–1% may face fatal outcomes [5]. The interpatient variability in pharmacokinetics is high and has been largely attributed to genetic polymorphisms affecting drug metabolism enzymes, particularly dihydropyrimidine dehydrogenase (DPD), encoded by the *DPYD* gene, which inactivates 80–90% of 5-FU [6].

Clinically validated *DPYD* variants such as *DPYD**2A, 13, c.2846A>T, and HapB3 have been strongly associated with reduced DPD activity and increased risk of FP-related toxicity. Heterozygous carriers of *DPYD**2A or 13 can experience a >60% enzyme reduction, while homozygous carriers may have complete DPD deficiency. Based on this, genotype-guided dose reduction is recommended by the Clinical Pharmacogenetics Implementation Consortium (CPIC) guidelines [8].

Recent evidence highlights that rare or population-specific *DPYD* variants may contribute significantly to toxicity risk. For instance, a large next generation sequencing (NGS) study in 2972 patients found that 77% of *DPYD* variants associated with DPD deficiency were rare (MAF < 1%), and over half were predicted to be deleterious. Importantly, many of these rare variants would be missed by conventional genotyping assays [9].

Beyond *DPYD*, other genes in the FP pathway—such as thymidylate synthase (*TYMS*), cytidine deaminase (*CDA*), and methylene tetrahydrofolate reductase (*MTHFR*)—have been implicated in treatment outcomes and toxicity. In a study by Loganayagam et al., the *TYMS* 3′-UTR del/del genotype was significantly associated with a threefold increase in overall FP toxicity (*p* = 0.01), while the *CDA* A79C polymorphism was associated with increased risk of neutropenia. Additionally, *MTHFR* c.1298A>C homozygosity showed a strong correlation with capecitabine-induced hand–foot syndrome [6].

NGS and other forms of genome-wide analyses have uncovered significant geographic and ethnic variations in genetic alterations associated with cancer, suggesting that drug efficacy and adverse effect profiles may differ across populations [10]. Effective prioritization and the implementation of drug–gene combinations for clinical testing in diverse ethnic groups necessitate an understanding of the distribution of relevant genetic variants and prevailing prescription patterns within each population. While pharmacogenomic screening is widely adopted in many medical institutions across the US and Europe, such implementation is lagging in many other parts of the world, including the Middle East [10].

The Sultanate of Oman is located in Southwest Asia, with a population of Arab descent. Smaller contributions to the population from individuals of Persian, South Asian, and African ancestries also exist. Oman’s unique demographics provide valuable opportunities for studying genetic diversity and understanding population-specific traits, which can inform research on personalized medicine and genetic health initiatives [11].

The aim of this study is to obtain a global profile of pharmacogenetic variants that are known to affect the FP pathway in healthy Omani individuals, forming a foundation for future studies in cancer patients undergoing FP-based treatments. We applied a targeted next generation sequencing panel to capture the coding and splice regions of 36 key genes involved in the FP pathway.

## 2. Materials and Methods

### 2.1. Participants

A total of 98 healthy participants who self-identified as Omanis were enrolled in this study. Recruitment was conducted at the Sultan Qaboos Comprehensive Cancer Care and Research Center, Muscat, Oman. All participants provided written informed consent prior to their inclusion in the study.

Whole-blood samples were collected in EDTA tubes from all participants, and DNA was subsequently extracted as described later. The inclusion criteria for participation were as follows:-Participants of Omani nationality, aged 18 years or older.-Individuals without any prior diagnosis of malignancy.-Completion of an informed consent form specific to healthy participants.

### 2.2. DNA Extraction

Whole-blood DNA samples were isolated with the MagMAX DNA Multi-Sample Ultra kit (Thermo Fisher Scientific, Waltham, MA, USA—Cat #: A36570) according to the manufacturer’s instructions. DNA quantity and quality were assessed using a Nanodrop 2000 spectrophotometer, a Qubit 3.0 fluorometer (Thermo Fisher Scientific, Waltham, MA, USA), and a Qubit dsDNA HS Assay Kit (Cat #: Q32851) to ensure precise quantification with sufficient DNA input for library preparation.

### 2.3. Targeted Gene Panel Next-Generation Sequencing

A custom gene panel was designed using the Ion AmpliSeq Designer (Thermo Fisher Scientific, Waltham, MA, USA) to capture the coding exons and splice-site regions of 36 genes involved in the metabolism of fluoropyrimidines. The genes included are as follows: *DPYD*, *DPYS*, *UPB1*, *CES1*, *CES2*, *UGT1A1*, *MTHFR*, *CYP2A6*, *ABCB1*, *ABCC3*, *ABCC4*, *ABCC5*, *ABCG2*, *SLC29A1*, *SLC22A7*, *TYMP*, *TP53*, *TK1*, *ERCC2*, *TYMS*, *UCK1*, *PPAT*, *GGH*, *UPP1*, *SMUG1*, *UMPS*, *UCK2*, *XRCC3*, *RRM1*, *TDG*, *CDA*, *RRM2*, *UPP2*, *DHFR*, *ENOSF1*, and *FPGS*. The corresponding protein products of these genes are provided in Appendix A. Library preparation was performed using the Ion AmpliSeq Library Kit (Thermo Fisher Scientific, Waltham, MA, USA Cat #: 4488990), and libraries were barcoded with the Ion Xpress Barcode Adapters 1–96 Kit (Thermo Fisher Scientific, Waltham, MA, USA Cat #: 4474517). Barcoded libraries were pooled to a final concentration of 100 pM. Lastly, libraries were purified using Agencourt™ AMPure™ XP beads (Beckman Coulter, Indianapolis, IN, USA—Cat #: A63880) and quantified using the Ion Library TaqMan™ Quantitation Kit (Thermo Fisher Scientific—Cat #: 4468802) on a QuantStudio 6 Flex™ Real-Time PCR System (Thermo Fisher Scientific, Waltham, MA, USA).

The Ion Chef™ System (Thermo Fisher Scientific, Waltham, MA, USA) was used for the preparation of the template and emulsion PCR with an Ion 520™ and Ion 530™ Kit—Chef (Cat #: A34019). Afterwards, the libraries were loaded onto Ion 530™ Chips (Cat #: A27763) and sequenced on the Ion S5™ System (Thermo Fisher Scientific, Waltham, MA, USA) using standard protocols.

Raw sequencing data were processed using Torrent Suite software (version 5.18) for base calling and alignment. Variants were annotated and analyzed using the Ion Reporter software (version 5.18) to identify clinically relevant polymorphisms and mutations. The clinical significance and drug sensitivity of the variants were verified using the ClinVar and dbSNP (NCBI, Bethesda, MD, USA), gnomAD (Broad Institute, Cambridge, MA, USA), and PharmGKB (Pharmacogenomics Knowledgebase, Stanford University, Stanford, CA, USA) databases.

## 3. Results

Ninety-eight healthy Omani volunteers participated in this group. The participants were 47 (48%) males and 51 (52%) females with an age range of 22–51 years.

Using targeted gene panel sequencing, a total of 999 distinct variants were detected across 36 genes using targeted gene panel sequencing, with these variants collectively occurring 12,947 times in the 98 healthy individuals. Notably, 63.3% of the genotypes were heterozygous, and the remaining were homozygous.

Among the different types of variants, single-nucleotide variants (SNVs) were the most prevalent, accounting for over 90%, followed by insertion–deletion (INDEL) variants, which constituted around 6%, and multi-nucleotide variants made up less than 1%.

It was found that the ABCC4 gene exhibited the highest number of mutations (76 mutations), followed by ABCB1 (67 mutations), CES1 (62 mutations), and CYP2A6 (61 mutations). In contrast, the genes DHFR and SMUG1 demonstrated the lowest mutation frequencies. This variability highlights ABCC4 as the most mutation-prone gene, while DHFR and SMUG1 appear to be the most stable (Figure 2).

The identified variants were cross-referenced with ClinVar to determine the clinical significance. Pathogenic, likely pathogenic, and variants with conflicting interpretations of pathogenicity are listed in Table 1. The MTHFR gene variant at locus chr1:11854476, identified by rs1801131, is a missense variant that changes a glutamic acid to an alanine (Glu429Ala). This variant is common in 63% of the population, with 44.9% having the genotype AC, and 18.4% had the genotype CC. The minor allele frequency (MAF) was 0.249, closely aligned with the global allele frequency (global MAF = 0.3032).

In the DPYD gene, four significant loci with functional impacts were identified. Within these sites, three showed missense alterations, identified as rs1801158 (Ser534Asn), rs45589337 (Pro86Leu), and rs568132506 (Arg78Ter), with corresponding occurrence rates in the population of 8.2%, 3.1%, and 1.0%, respectively. Furthermore, one locus, rs776692894, demonstrated a nonsense mutation, observed in 2.0% of the population.

The UGT1A1 gene also featured a missense variant, rs35003977, found in 1% of the population. Furthermore, in the UPB1 gene, a mutation at locus chr22:24896073 (rs138081800) involving a substitution of adenine (A) to guanine (G) was observed in 4.1% of the population. Notably, most MAFs of the variants in our study population were comparable to the Global MAF data from the gnomeAD database, as shown in Table 1.

Among the investigated gene variants, three significant variants in two genes—*DPYD* and *MTHFR*—were identified as been associated with the drug response to 5-FU and/or capecitabine, as shown in Table 2 [12,13,14]. These findings underscore the potential influence of these variants on drug toxicity and efficacy within the Omani population. For each of these significant variants, the frequency was relatively low, with occurrence rates identified in only 1% of the cohort. The MAF of the identified variants within the Omani population were compared to the global MAFs, as summarized in Table 2. Most variants showed no substantial differences between the observed allele frequencies in our population and the corresponding GMAF values. However, the *DPYD* c.557A>G (rs115232898) variant exhibited a notably higher MAF in our population (0.006) compared to the global frequency (GMAF = 0.001076).

In addition to the clinically relevant variants, a total of 27 unique SNVs were detected and are detailed in Table 3. These comprised 3 synonymous variants, 5 missense variants, 1 nonsense variant, and 18 variants of unknown significance. Notably, the most common unique variant was identified in the *ABCC3* gene, observed in 43.9% of the population. The functional implications of these novel variants remain to be elucidated.

## 4. Discussion

This study represents the first targeted pharmacogenomic profiling of 36 genes associated with chemotherapy outcomes in 98 healthy Omani volunteers. It identified significant variants in genes associated with chemotherapy drug responses, such as *DPYD* and *MTHFR*. Additionally, novel variants in the *ABCC3* gene were identified that require further investigation. These findings provide critical insights into the genetic variability of the Omani population.

We utilized a targeted gene panel focused on genes of interest, carefully selected based on prior studies investigating the impact of genetic mutations within the FP metabolic pathway [4]. The decision to use a targeted gene panel was particularly well suited for this investigation, as it enabled a focused analysis of genes with well-established roles in chemotherapy metabolism, efficacy, and toxicity. Given that this is the first study to explore these genetic variants in the Omani population, our approach maximized the clinical and research relevance while minimizing unnecessary complexity and resource use [15]. Furthermore, the findings from this study serve as a pivotal bridge to clinical studies involving cancer patients undergoing chemotherapy affected by these genes.

Our data showed that most of the identified genotypes (63.3%) were heterozygous, indicating that participants carried two different alleles for these genes. This reflects a significant degree of genetic diversity probably due to a limited set of common variants, even in a healthy population [16].

Moreover, the data revealed that single-nucleotide variants (SNVs) are the dominant type of genetic variation, which compromises over 90% of the variants observed. This is consistent with global genomic studies, showing a high occurrence rate of SNVs, which is about 1 per 800 bases in humans [16].

Variants in ABC transporter genes, including *ABCB1*, *ABCC4*, and *ABCC3*, as well as solute carrier (SLC) genes such as *SLC28A3*, are known to influence the pharmacokinetics of numerous anti-cancer agents, including fluoropyrimidines, methotrexate, anthracyclines, and irinotecan. Specifically, *ABCC4* encodes an efflux transporter involved in the clearance of nucleotide analogs and active fluoropyrimidine metabolites like FUMP and FdUMP, modulating intracellular drug concentrations and thereby impacting both toxicity and treatment response. Although *ABCB1* does not directly efflux 5-FU, its overexpression has been associated with treatment resistance in clinical samples, possibly due to interactions with other transporter and metabolic pathways [17,18].

The *ABCC4* gene exhibited the highest mutation count in our study, with 76 variants, including 1 novel intronic variant. This high variability may reflect both the gene’s large size and its susceptibility to replication errors. Notably, *ABCC4* is highly expressed in tissues exposed to xenobiotics and during chemically induced stress, providing a permissive environment for DNA damage and potential mutagenesis [18]. In addition to *ABCC4*, other ABC transporter genes such as *ABCB1* (67 variants) and *ABCC3* also showed significant variation, including a novel *ABCC3* insertion (c.806+39_806+40insT) found in 43.9% of the population—suggesting a possible population-specific polymorphism.

Emerging pharmacogenetic evidence supports the clinical relevance of ABC transporter variants. For instance, *ABCB1* rs1045642 and *ABCC11* rs7194667 have been associated with an increased risk of fluoropyrimidine-induced adverse events, including diarrhea, mucositis, and leukopenia [17,18].

The *UPB1* variant rs138081800, located at the critical splice acceptor site of intron 1 (c.105-2A>G), was observed in 4.1% of healthy Omani individuals. While the precise functional impact of this variant remains to be fully characterized, its location at a splice site suggests a potential influence on mRNA processing or transcript stability [19]. The *UPB1* gene encodes β-ureidopropionase, a key enzyme in the terminal step of pyrimidine catabolism, catalyzing the conversion of N-carbamyl-β-alanine to β-alanine. Disruption in this pathway has been implicated in uraciluria and neurological disorders, and more recently, its relevance to the metabolism of FP drugs has been recognized [20].

Although pharmacogenomic data on rs138081800 specifically is limited, *UPB1* mutations may affect pyrimidine degradation kinetics and thus alter drug levels or toxicity profiles for FP-based therapies. In particular, *UPB1* dysfunction could lead to the accumulation of toxic metabolites, paralleling mechanisms observed in DPYD deficiency [19,20].

Several *DPYD* gene variants were identified in our cohort of healthy Omani volunteers, including *rs1801158*, *rs45589337*, *rs568132506*, *rs776692894*, *rs3918290*, and *rs115232898*. Among these, *rs568132506* and *rs776692894* (missense and nonsense variants, respectively) are classified as pathogenic or likely pathogenic, and are associated with significantly reduced DPD activity—the key enzyme responsible for the catabolism of 5-FU [21]. Similarly, *rs3918290*, a well-characterized splice-site variant, results in a truncated, nonfunctional protein and is associated with complete loss of DPD activity [22,23,24]. Missense variant *rs115232898* also leads to reduced enzymatic function and has been linked to an increased risk of fluoropyrimidine-related toxicity [25,26]. Collectively, these four variants are clinically significant, as they predispose carriers to potentially life-threatening toxicities such as myelosuppression and mucositis upon treatment with 5-FU or capecitabine. The higher observed frequency of the *DPYD* c.557A>G (rs115232898) variant in our cohort compared to the GMAF may indicate a population-specific enrichment. Additionally, *rs1801158*, a missense variant of uncertain functional significance, was found at a frequency of 8.2% in our population—markedly higher than the 0.5% reported in European populations [27]. This notable prevalence difference may have implications for regional pharmacogenomic screening policies. The detection of multiple clinically relevant *DPYD* variants in our study supports the recommendation for routine *DPYD* genotyping prior to the initiation of FP-based therapy to guide dosing and improve patient safety.

The *MTHFR* variant *rs1801131* (A1298C) was prevalent in 63% of participants, with genotypes AC (44.9%) and CC (18.4%). By impairing folate metabolism, it was shown that both the A1298C, rs1801131 and the less common variant C677T can lead to a reduction in enzymatic activity, influencing the efficacy and toxicity of antifolate chemotherapeutics [28]. Additionally, the C677T variant (*rs1801133*), though it was only found in 1% of our population, is well-documented for its association with increased 5-FU toxicity in colorectal cancer patients [29,30].

The *UGT1A1* missense variant *rs35003977*, identified in 1% of participants, is associated with irinotecan-induced toxicity [24]. In our study, we identified the *UGT1A1* missense variant *rs35003977* (p.Val225Gly), present in 1% of the population with an MAF of zero, which aligns with findings from Oussalah C et al., 2015, who reported this low-frequency, rare variant in the substrate-binding domain coding region of exon 1 in *UGT1A1*, potentially altering the binding site due to its proximity to loop 5 on the *UGT1A1* protein [31].

The observed MAFs of the identified variants largely align with global data from the gnomAD database, reflecting consistency in genetic variability. However, regional differences underscore the need for population-specific research. For example, we identified several variants with differing frequencies compared to global databases such as gnomAD. For instance, the *MTHFR* variant c.1286A>C (rs1801131) had a notably lower MAF in our population (0.249) than the global average (GMAF = 0.3032), while *MTHFR* c.665C>T (rs1801133), another common functional polymorphism, showed a comparable MAF (0.245) to global reports (GMAF = 0.3182). The observed prevalence also aligns with findings from Middle Eastern populations [32], where the AC and CC genotypes of the rs1801131 SNP were found in 38% and 19% of the Iranian population, respectively.

Conversely, some *DPYD* variants such as c.257C>T (rs568132506) and c.232A>T (rs776692894), classified as pathogenic or likely pathogenic, were found in our population at frequencies (1–2%) much higher than those observed globally (<0.0001), underscoring the importance of population-specific screening in fluoropyrimidine pharmacogenetics. Moreover, the higher observed frequency of the *DPYD* c.557A>G (rs115232898) variant in our cohort compared to the GMAF may indicate a population-specific enrichment. Additionally, rs1801158, a missense variant of uncertain functional significance, was found at a frequency of 8.2% in our population—markedly higher than the 0.5% reported in European populations [27]. This notable prevalence difference may have implications for regional pharmacogenomic screening policies. It is worth noting that these findings are consistent with studies from neighboring regions, such as Saudi Arabia, which report similar trends in *DPYD* and *MTHFR* polymorphisms [33].

Furthermore, a comparison with gnomAD African, East Asian, and European populations showed that some variants, while not entirely novel, had markedly different frequencies. For example, *UPB1* rs138081800 was observed in 4.1% of our population versus 0.4% globally, with slightly higher frequencies in East Asian populations (around 1%) and much lower frequencies in African and European populations (often < 0.1%). This reinforces the possibility of regional enrichment due to founder effects or local selection pressures [20].

We also detected 27 variants absent or extremely rare in international databases, including gnomAD and dbSNP, and not previously annotated in ClinVar. These were classified as novel, with the most frequent being *ABCC3* c.3251C>T (p.Ala1084Val), observed in 43.9% of the population. This particular variant has not been reported in any global dataset, suggesting a potentially unique allele in the Omani population. The predominance of such variants reflects the under-representation of Middle Eastern populations in genomic databases and highlights the added value of conducting pharmacogenetic studies in this context.

These findings collectively emphasize the need for population-specific pharmacogenomic profiling, particularly for the implementation of genotype-guided chemotherapy and for ensuring equitable access to precision medicine.

Additional rare but clinically relevant variants were found in the genes *UGT1A1* and *UPB1*.

It is important to highlight that over 60% of the identified variants were rare, a finding consistently reported in population studies analyzing pharmacogenes through NGS [34,35]. Notably, a substantial proportion of these rare variants were novel and absent from public genetic databases. Rare variants are known to occur frequently (one variant every 17 bases) and are often private to specific populations [36]. Recent evidence suggests that rare variants may account for 30% to 40% of the variability in drug response [37]. Furthermore, characterizing the rare and population-specific pharmacogenomic polymorphisms would emphasize the limitations of widely used genotyping platforms, which often suffer from the insufficient coverage of global populations [38].

There are a few limitations of the current study. For example, a relatively small number of samples were sequenced. However, for pharmacogenomic studies focusing on rare and novel variants, small cohorts can still yield meaningful insights, especially when combined with high-depth sequencing. Moreover, such exploratory studies are critical for laying the groundwork for larger-scale population studies [39,40]. Also, the genetic sub-structure of the Omani population was not considered in the current study. However, although the population of Oman is described as highly admixed, the population’s genetic sub-structure has never been described comprehensively in the literature. Finally, the sequencing panel largely targeted the coding regions of the genes, whereas some of the actionable variants lay in the intronic or untranslated regions.

## 5. Conclusions

In conclusion, our study emphasizes the importance of targeted gene panel sequencing in understanding genetic variation in a healthy population, as it enables the identification of variants that may serve as a baseline for future comparisons in studies focused on drug-related mutations.

## Figures and Tables

**Figure 1 genes-16-00592-f001:**
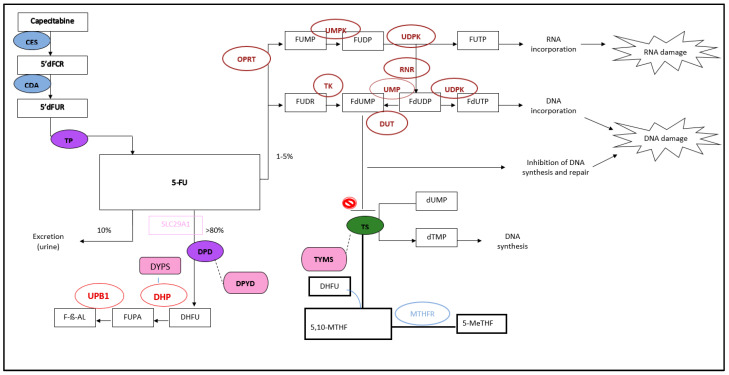
Metabolism of CAP and 5-FU [6,7]. Enzymes: carboxyl esterase (CES), cytidine deaminase (CDA), deoxyuridine triphosphatase (DUT), dihydropyrimidinas (DHP), dihydropyrimidine dehydrogenase (DPD), methylene tetrahydrofolate reductase (MTHFR), orotate phosphoribosyltransferase (OPRT), ribonucleotide reductase (RNR), thymidine kinase (TK), thymidine phosphorylase (TP), thymidine synthase (TS), β-ureidopropionase (UPB1), uridine diphosphate kinase (UDPK), uridine kinase (UK), and uridine monophosphate kinase (UMPK). Metabolites: deoxyfluorocytidine riboside (5′-dFCR), deoxyfluorocytidine monophosphate (5′-dFCMP), deoxyfluorouridine monophosphate (5-FdUMP), deoxyfluorouracil (5′-dFUR), fluorouracil (5-FU), fluorouridine (5-FUridine), fluorouracil monophosphate (5-FUMP), fluorouracil di, tri-phosphate (5-FUDP, 5-FUTP), deoxyfluorouracil di, tri-phosphate (5-FdUDP, 5-FdUTP), deoxyuridine mono, tri-phosphate (dUMP, dUTP), deoxycytidine mono, tri-phosphate (dTMP, dTTP), 5,10-methylenetetrahydrofolate (5,10-MTHF), 5-methyltetrahydrofolate (5-MeTHF), dihydrofolate (DHF), dihydrofluorouracil (DHFU), beta-fluoroureido propionic acid (b-FUPA), and fluoro-b-alanin (F-b-AL). Genes: dihydropyrimidinase (*DPYS*), dihydropyrimidine dehydrogenase (*DPYD*), and thymidylate synthase (*TYMS*).

**Figure 2 genes-16-00592-f002:**
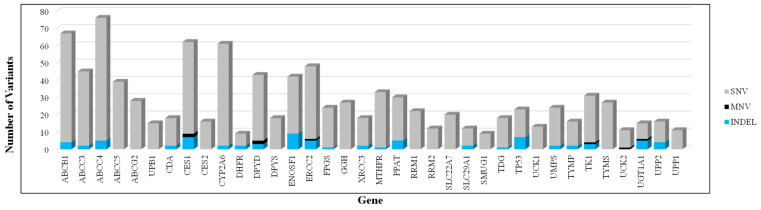
Distribution of genetic variants across pharmacogenes relevant to the fluoropyrimidine pathway. Bar chart depicting the total number and classification of genetic variants identified in 36 pharmacogenes associated with drug metabolism, transport, or response. Each bar corresponds to a single gene and reflects the cumulative number of variants detected in that gene across all study participants. Variants are categorized into three types: single-nucleotide variants (SNVs, shown in grey), multi-nucleotide variants (MNVs, shown in black), and insertions–deletions (INDELs, shown in blue).

**Table 1 genes-16-00592-t001:** Chromosomal loci associated with genes and their pathogenic phenotypes.

Genes	Locus	Variant	Function (Variant Type)	Genotype	dbSNP	Clinical Significance in Clinvar	% of Population	MAF	GMAF from Genome AD
*MTHFR*	chr1:11854476	NM_005957.5:c.1286A>C:p.Glu429Ala	missense	AC	rs1801131	CP	44.9	0.249	0.3032
	chr1:11854476	NM_005957.5:c.1286A>C:p.Glu429Ala	missense	CC	rs1801131	CP	18.4		
*DPYD*	chr1:97981421	NM_000110.4:c.1601G>A:p.Ser534Asn	missense	AG	rs1801158	CP	8.2	0.01	0.01878
	chr1:98144726	NM_000110.4:c.775A>G:p.Lys259Glu	missense	AG	rs45589337	CP	3.1	0.004	0.006651
	chr1:98206012	NM_000110.4:c.257C>T:p.Pro86Leu	missense	CT	rs568132506	P, LP	1	0.000	0.0000806
	chr1:98293671	NM_000110.4:c.232A>T:p.Arg78Ter	nonsense	AT	rs776692894	LP	2	NA	0.000003099
*UGT1A1*	chr2:234669607	NM_000463.3:c.674T>G:p.Val225Gly	missense	TG	rs35003977	P, LP, CP, US	1	0.000	0.000485
*UPB1*	chr22:24896073	NM_016327.3:c.105-2A>G:p.?	splice site	AG	rs138081800	CP	4.1	NA	0.000198

dbSNP: Database of Single-Nucleotide Polymorphisms; MAF: Minimum Allele Frequency; GMAF: Global Minimum Allele Frequency; AD: Aggregation Database; CP: conflicting interpretations of pathogenicity; P: pathogenic; LP: likely pathogenic; NA: not available; US: uncertain significance.

**Table 2 genes-16-00592-t002:** List of drug-related variants.

Genes	Locus	Variant	Function (Variant Type)	Genotype	dbSNP	% from Population	MAF	GMAF
*DPYD*	chr1:97915614	NM_000110.4:c.1905+1G>A	unknown	G/A	rs3918290	1	0.003	0.005061
	chr1:98165030	NM_000110.4: c.557A>G:p.Tyr186Cys	missense	A/G	rs115232898	1	0.006	0.001076
*MTHFR*	chr1:11856378	NM_005957.5:c.665C>T:p.Ala222Val	missense	C/T	rs1801133	1	0.245	0.3182

dbSNP; Database of Single-Nucleotide Polymorphisms, MAF; Minimum Allele Frequency, GMAF; Global Minimum Allele Frequency.

**Table 3 genes-16-00592-t003:** List of Novel Variants.

Genes	Locus	Variant	Function (Variant Type)	Genotype	% from Population
*ABCB1*	chr7:87232389	NM_000927.4:c.1838A>G:p.Asp613Gly	unknown	C/T	1
*ABCC3*	chr17:48736768	NM_003786.4:c.806+39_806 + 40insT	unknown	C/CT	1
	chr17:48742499	NM_003786.4:c.1339-15C>T	unknown	C/T	1
	chr17:48745355	NM_003786.4:c.1767C>T:p.Ile589=	synonymous	C/T	1
	chr17:48753822	NM_003786.4:c.3251C>T:p.Ala1084Val	missense	C/T	43.9
	chr17:48762162	NM_003786.4:c.4206C>G:p.Ser1402=	synonymous	C/G	1
*ABCC4*	chr13:95815936	NM_005845.5:c.2176-35T>C	unknown	A/G	1
*ABCG2*	chr4:89028449	NM_004827.3:c.1195-31C>T	unknown	G/A	1
*CES1*	chr16:55844963	NM_001025195.2:c.1087-41T>C	unknown	A/G	1
*DPYD*	chr1:97847979	NM_000110.4:c.1942_1943delAA:p.Asn648Ter	nonsense	ATT/A	1
*ENOSF1*	chr18:691073	NM_017512.7:c.530A>G:p.Glu177Gly	missense	T/C	1
	chr18:712705	NM_017512.7:c.-136T>CT	unknown	A/AG	1
	chr18:712723	NM_017512.7:c.-170G>C	unknown	A/AG	1
	chr18:712757	NM_017512.7:c.-118T>CT	unknown	C/G	1
*ERCC2*	chr19:45855067	NM_000400.4:c.2191-88C>T	unknown	G/A	1
	chr19:45873811	NM_000400.4:c.-13GT>T	unknown	AC/A	1
*FPGS*	chr9:130569681	NM_004957.6:c.580-19G>A	unknown	G/A	1
*RRM1*	chr11:4156257	NM_001033.5:c.2002-55C>G	unknown	C/G	1
*SLC22A7*	chr6:43266286	NM_153320.2:c.190T>G:p.Trp64Gly	missense	T/G	2
*TK1*	chr17:76182814	NM_003258.5:c.98+53delC	unknown	AG/A	1
	chr17:76182814	NM_003258.5:c.98+53delC	unknown	A/A	2
*TYMS*	chr18:671311	NM_001071.4:c.733-69A>C	unknown	A/C	3
*PPAT*	chr4:57272852	NM_002703.5:c.211A>G:p.Asn71Asp	missense	T/C	1
	chr4:57301450	NM_002703.5:c.128+66G>C	missense	C/G	1
*UCK2*	chr1:165797127	NM_012474.5:c.57G>A	synonymous	G/A	1
*UGT1A1*	chr2:234676396	NM_019076.4:c.988-99_988-98insGT	unknown	T/TGT	3
*XRCC3*	chr14:104173311	NM_001100118.2:c.406 + 29G>A	unknown	C/T	1

## Data Availability

Protocols and de-identified, aggregated data that underlie the results reported in this article are available for non-commercial scientific purposes upon reasonable request from the corresponding author.

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
