# Peer review of "Pharmacogenetic Profiling of Genes Associated with Outcomes of Chemotherapy in Omani Healthy Controls"

_genes, 2025, doi:10.3390/genes16050592_

Round 1
Reviewer 1 Report
Comments and Suggestions for Authors
The authors conducted a study on the fluoropyrimidine-related gene landscape in Omani healthy population. Drug-related variants as well as novel variants have been identified. This descriptive study's findings might be relevant for further studies on disease-related mutations in the given population.
Several points for authors' consideration:
In the methods part please provide the catalog numbers for all the products to facilitate the reproducibility of your research.
Please provide a more detailed Figure 1 legend describing what exactly is depicted.
Lines 167-169 - are there literature sources confirming this? If so, please provide citations.
Author Response
In the methods part please provide the catalog numbers for all the products to facilitate the reproducibility of your research.
We thank the reviewer for this valuable suggestion. We have now included the catalog numbers for all the kits and reagents used in the study to enhance reproducibility. The following catalog numbers have been added to the revised Methods section:
- MagMAX™ DNA Multi-Sample Ultra Kit (Cat #: A36570)
- Qubit™ dsDNA HS Assay Kit (Cat #: Q32851)
- Ion AmpliSeq™ Library Kit (Cat #: 4488990)
- Ion Xpress™ Barcode Adapters 1–96 Kit (Cat #: 4474517)
- Agencourt™ AMPure™ XP beads (Cat #: A63880)
- Ion Library TaqMan™ Quantitation Kit (Cat #: 4468802)
- Ion 520™ & Ion 530™ Kit (Cat #: A34019)
- Ion 530™ Chips (Cat #: A27763)
Please provide a more detailed Figure 1 legend describing what exactly is depicted.
We thank the reviewer for this helpful comment. We apologize for the oversight in the previous version of the figure legend, which did not clearly define the abbreviations. We have now revised the legend to provide a more comprehensive explanation of the data and to clarify all abbreviations used. Also, because we have added a figure in the introduction, this figure has now become figure 2. The updated legend is as follows:
Figure 2. Distribution of genetic variants across pharmacogenes relevant to fluoropyrimidine pathway.
Bar chart depicting the total number and classification of genetic variants identified in 36 pharmacogenes associated with drug metabolism, transport, or response. Each bar corresponds to a single gene and reflects the cumulative number of variants detected in that gene across all study participants. Variants are categorized into three types: single nucleotide variants (SNVs, shown in grey), multi-nucleotide variants (MNVs, shown in black), and insertions/deletions (INDELs, shown in blue).
Lines 167-169 - are there literature sources confirming this? If so, please provide citations
Yes, relevant literature sources are available to support this statement, and we have added them as citations in the revised manuscript as suggested. The newly added references are now listed as references 12-14.
Reviewer 2 Report
Comments and Suggestions for Authors
In this paper, the authors screened healthy Omani individuals for pharmacogenetic variants in 36 genes related to the fluoropyrimidine (FP) pathway. A high percentage of modifications in the ABCC4 gene was observed and lower percentage for other genes. Importantly, 75 novel SNPs were detected. Although the number of participants is not very high in this report (only 98 individuals), the results of this study are significant for personalized medicine.
The paper is well-written, some minor adjustment required.
Line 43: Evaluate the addition of the most important modified genes to the Keywords section.
Introduction: Reference [1] reports data from 2020, while the webpage [1] provides data from 2022. Please update the Introduction accordingly.
Methods, Line 110: When listing all studied genes, also include the names of their corresponding protein products. This will facilitate the understanding of potential disruptions in protein function.
Line 132: “A total” – is the uppercase “A” a typo?
Figure 1: The abbreviations SNV, MNV, and INDEL used in the figure should be mentioned and explained in the figure legend.
Table 1: In the “Clinical Significance” column for MTHFR, the term “other” is used. What does “other” refer to? Clarify this in the footnotes.
Lines 167-169: After the statement “Among the investigated...,” include the citation.
Line 173: The authors state, “There were no substantial differences…” in Table 2. This is correct for two of the three reported variants. However, DPYD A/G shows a MAF of 0.006 compared to a GMAF of 0.001076, which constitutes a substantial difference. Please highlight this in the results and discuss or comment on it in the Discussion section.
Line 179: The authors claim that “75 unique SNVs were detected and detailed in Table 3.” However, Table 3 reports only 27 variants. Please correct this discrepancy.
Line 307, Abbreviations: Numerous abbreviations used in the paper are missing (as gene names). Carefully review and ensure that all abbreviations are listed.
Suggestion for the further study:
It would be both interesting and important to test patients—specifically individuals diagnosed with cancer—by analyzing their genotype using a sample collected immediately upon diagnosis. This approach would better align with the genetic profile of the individuals who will effectively receive therapy.
Author Response
Line 43: Evaluate the addition of the most important modified genes to the Keywords section.
Thank you for the valuable suggestion. We have revised the Keywords section to include the most relevant and frequently discussed pharmacogenes in our study, including DPYD, MTHFR, and ABCC4 and UPB1 genes.
Introduction: Reference [1] reports data from 2020, while the webpage [1] provides data from 2022. Please update the Introduction accordingly.
We thank the reviewer for this observation. We have updated the Introduction to reflect the most recent global cancer incidence data from 2022, in alignment with the cited reference.
Revised Sentence in the Manuscript:
The World Cancer Research Fund reports that more than 18 million new cases were diagnosed globally in 2022 [1].
Methods, Line 110: When listing all studied genes, also include the names of their corresponding protein products. This will facilitate the understanding of potential disruptions in protein function.
We thank the reviewer for this helpful recommendation. To improve clarity and support the interpretation of functional consequences, we have now provided the full list of all 36 pharmacogenes analyzed in the study along with their corresponding protein products. This information has been added as Appendix 1 in the revised manuscript.
Gene |
Protein Product |
DPYD |
Dihydropyrimidine dehydrogenase |
DPYS |
Dihydropyrimidinase |
UPB1 |
β-ureidopropionase 1 |
CES1 |
Carboxylesterase 1 |
CES2 |
Carboxylesterase 2 |
UGT1A1 |
UDP-glucuronosyltransferase 1A1 |
MTHFR |
Methylenetetrahydrofolate reductase |
CYP2A6 |
Cytochrome P450 2A6 |
ABCB1 |
ATP-binding cassette transporter subfamily B1 |
ABCC3 |
ATP-binding cassette transporter subfamily C3 also known as Multidrug resistance-associated protein 3 (MRP3) |
ABCC4 |
ATP-binding cassette transporter subfamily C4 also known as Multidrug resistance-associated protein 4 (MRP4) |
ABCC5 |
ATP-binding cassette transporter subfamily C5 also known as Multidrug resistance-associated protein 5 (MRP5) |
ABCG2 |
ATP-binding cassette transporter subfamily G2 also known as Breast cancer resistance protein (BCRP) |
SLC29A1 |
Solute carrier family 29 member A1 also known as Equilibrative nucleoside transporter 1 (ENT1) |
SLC22A7 |
Solute carrier family 22 member A7 also known as Organic anion transporter 2 (OAT2) |
TYMP |
Thymidine phosphorylase |
TP53 |
Cellular tumor antigen p53 |
TK1 |
Thymidine kinase 1 |
ERCC2 |
DNA excision repair protein ERCC-2 (XPD) |
TYMS |
Thymidylate synthase |
UCK1 |
Uridine-cytidine kinase 1 |
PPAT |
Phosphoribosyl pyrophosphate amidotransferase |
GGH |
Gamma-glutamyl hydrolase |
UPP1 |
Uridine phosphorylase 1 |
SMUG1 |
Single-strand-selective monofunctional uracil-DNA glycosylase |
UMPS |
Uridine monophosphate synthetase |
UCK2 |
Uridine-cytidine kinase 2 |
XRCC3 |
X-Ray Repair Cross Complementing 3 also known as DNA repair protein XRCC3 |
RRM1 |
Ribonucleoside-diphosphate reductase subunit M1 |
TDG |
Thymine DNA glycosylase |
CDA |
Cytidine deaminase |
RRM2 |
Ribonucleoside-diphosphate reductase subunit M2 |
UPP2 |
Uridine phosphorylase 2 |
DHFR |
Dihydrofolate reductase |
ENOSF1 |
Enolase superfamily member 1 |
FPGS |
Folylpolyglutamate synthase |
Line 132: “A total” – is the uppercase “A” a typo?
We thank the reviewer for pointing this out. Yes, this was a typographical error, and we have now corrected “A total” to lowercase in the revised manuscript (line 171 in the revised paper).
Figure 1: The abbreviations SNV, MNV, and INDEL used in the figure should be mentioned and explained in the figure legend.
We thank the reviewer for this helpful comment. We apologize for the oversight in the previous version of the figure legend, which did not clearly define the abbreviations or fully describe the contents of the figure. We have now revised the legend to provide a more comprehensive explanation of the data and to clarify all abbreviations used. The updated legend is as follows:
Figure 2. Distribution of genetic variants across pharmacogenes relevant fluoropyrimidine pathway.
Bar chart depicting the total number and classification of genetic variants identified in 36 pharmacogenes associated with drug metabolism, transport, or response. Each bar corresponds to a single gene and reflects the cumulative number of variants detected in that gene across all study participants. Variants are categorized into three types: single nucleotide variants (SNVs, shown in grey), multi-nucleotide variants (MNVs, shown in black), and insertions/deletions (INDELs, shown in blue).
Table 1: In the “Clinical Significance” column for MTHFR, the term “other” is used. What does “other” refer to? Clarify this in the footnotes.
We thank the reviewer for this useful comment. To avoid ambiguity, we have removed the term “other” from the Clinical Significance column for both MTHFR and UGT1A1.
- For MTHFR, the term “other” has been deleted entirely.
- For UGT1A1, “other” has been replaced with the specific classifications: P (Pathogenic), LP (Likely Pathogenic), US (Uncertain Significance). The existing designation CP (conflicting interpretation of pathogenicity) has been retained.
- Additionally, we have added a clarifying footnote to the table:
US: Uncertain Significance
These edits have been made in Table 1 and its corresponding footnotes in the revised manuscript.
Lines 167-169: After the statement “Among the investigated...,” include the citation.
Included now; references 12-14.
Line 173: The authors state, “There were no substantial differences…” in Table 2. This is correct for two of the three reported variants. However, DPYD A/G shows a MAF of 0.006 compared to a GMAF of 0.001076, which constitutes a substantial difference. Please highlight this in the results and discuss or comment on it in the Discussion section.
We thank the reviewer for this important observation. We agree the suggested change. We have revised the Results section accordingly to clarify this point, and have also added a corresponding comment in the Discussion section.
Revised text in Results (Lines 220-224):
Most variants showed no substantial differences between the observed allele frequencies in our population and the corresponding GMAF values. However, the DPYD c.557A>G (rs115232898) variant exhibited a notably higher MAF in our population (0.006) compared to the global frequency (GMAF = 0.001076).
Line 179: The authors claim that “75 unique SNVs were detected and detailed in Table 3.” However, Table 3 reports only 27 variants. Please correct this discrepancy.
We thank the reviewer for pointing out this discrepancy. The number “75” corresponded to the total count of the novel 27 SNVs among 98 participants. We have corrected the statement to reflect the accurate number of new variants detailed in Table 3. The revised paragraph is provided below:
Revised text (Lines 228-230):
“In addition to the clinically relevant variants, a total of 27 unique SNVs were detected and are detailed in Table 3. These comprised 3 synonymous variants, 5 missense variants, 1 nonsense variant, and 18 variants of unknown significance. Notably, the most common unique variant was identified in the ABCC3 gene, observed in 43.9% of the population. The functional implications of these novel variants remain to be elucidated.”
Line 307, Abbreviations: Numerous abbreviations used in the paper are missing (as gene names). Carefully review and ensure that all abbreviations are listed.
Thank you for highlighting this oversight. We have now carefully reviewed the manuscript and ensured that all abbreviations, including gene names and pathway-related terms, are listed in the Abbreviations section.
Suggestion for the further study:
It would be both interesting and important to test patients—specifically individuals diagnosed with cancer—by analyzing their genotype using a sample collected immediately upon diagnosis. This approach would better align with the genetic profile of the individuals who will effectively receive therapy.
We thank the reviewer for this insightful suggestion. We agree that analyzing the pharmacogenomic profile of patients at the time of cancer diagnosis would provide direct clinical relevance and help tailor therapy more effectively. In fact, our current study represents an initial exploratory phase focused on healthy individuals to establish a population-level baseline of pharmacogenomic variants in the Omani population. We are currently planning a subsequent phase of this research, which will involve genotyping cancer patients who are planned for chemotherapy, as these are germline variants. This will allow us to evaluate the clinical impact of these variants on treatment outcomes and toxicity. We believe that the present findings lay important groundwork for such future investigations.
Reviewer 3 Report
Comments and Suggestions for Authors
Thank you very much for providing this principle interesting study.
The aim to identify interesting target SNPs for cancer seems to be relevant. However, I do not understand, why only healthy individuals were included into the study? There should be included persons reciving chemotherapy. In addtion, as in pricple it seems relevant to detect polymorphisms in this special gene set, I wonder if the identified polymorphisms are really new? There should be made a comparision, to other ethic groups. As some of the SNPs alrerady have rs -Numbers there seem to be known before.
A third point that seems to be positive is that the genes selected seems to be relevant in the described pathway, which makes it a specilized study.
Hoewever the sample size of the study seems very low for genetic detection and the identified variants were not functionally tested,. This should be carried out for at least the most relevant variants.
Author Response
Comments and Suggestions for Authors
Thank you very much for providing this principle interesting study.
The aim to identify interesting target SNPs for cancer seems to be relevant. However, I do not understand, why only healthy individuals were included into the study? There should be included persons receiving chemotherapy. In addition, as in principle it seems relevant to detect polymorphisms in this special gene set, I wonder if the identified polymorphisms are really new? There should be made a comparison, to other ethnic groups. As some of the SNPs already have rs -Numbers there seem to be known before.
A third point that seems to be positive is that the genes selected seems to be relevant in the described pathway, which makes it a specialized study.
However, the sample size of the study seems very low for genetic detection and the identified variants were not functionally tested. This should be carried out for at least the most relevant variants.
We sincerely thank the reviewer for their valuable and thoughtful feedback. We appreciate the recognition of the relevance of the target genes selected and the potential of this study to contribute to pharmacogenetic understanding in the context of chemotherapy.
Regarding the inclusion of healthy individuals, the current study was intentionally designed to characterize the baseline distribution of pharmacogenetic variants in the Omani population. Establishing baseline data in unaffected individuals allows for the identification of background variant frequencies without confounding factors such as malignancy-related genetic changes, chemotherapy exposure, or treatment-induced selection pressures. This provides a necessary reference point for future investigations involving cancer patients receiving chemotherapy, which are currently being planned as an extension of this research.
Concerning the novelty of the detected variants, we acknowledge that many variants have already been annotated with rs numbers. The aim of the study was not primarily to identify novel variants, but rather to provide the first population-specific profiling of these pharmacogenes in the Omani population, where such data has been previously lacking. This profiling is critical for informing the prioritization of clinically actionable variants in subsequent applied and translational research.
In response to the reviewer’s valuable comment, we have now expanded the discussion section to include a comparative analysis of allele frequencies against available data from other ethnic groups, utilizing public resources such as gnomAD (lines 324-360).
We agree that the relatively modest sample size is a limitation for detecting rare variants with high confidence. However, it is sufficient for initial population characterization and frequency estimation of common and moderately rare variants, which was the primary objective of this stage. Larger sample sizes, including disease-specific cohorts, are being considered for future studies, such as the Omani national genome project.
Finally, we recognize that functional validation of the detected variants was not performed in this study. Functional studies focusing on selected high-priority variants are part of the planned future work to build on the findings reported here.
Round 2
Reviewer 3 Report
Comments and Suggestions for Authors
Thank you very much for providing additional data of the manuscript, which largerly improved it. Especially the discussion now states all aspects, I was missing before.